# Diagnostic and Management Challenges of Esophageal Rupture with Concomitant Cervical Abscess in Chronic High Cervical Tetraplegia

**DOI:** 10.3390/diagnostics14040391

**Published:** 2024-02-11

**Authors:** Junghwan Park, Dong Gyu Lee

**Affiliations:** Department of Physical Medicine and Rehabilitation, Yeungnam University College of Medicine, Daegu 42415, Republic of Korea; ddi2019@gmail.com

**Keywords:** esophageal rupture, tetraplegia, cervical fusion, pyogenic spondylitis

## Abstract

A 65-year-old with a history of spinal cord injury and previous cervical surgery presented with persistent fever despite antibiotic treatment. MRI scans revealed an abscess in the neck extending from C3 to C6, with associated osteomyelitis. After an initial discharge following antibiotic therapy, the patient was readmitted due to recurrent systemic infection symptoms and another abscess. A subsequent endoscopy showed esophageal rupture with protruding cervical fusion metal. Due to operative risks, a percutaneous endoscopic gastrostomy was performed without further infection recurrence. The absence of typical imaging signs of esophageal rupture made diagnosis difficult. The infection spread through the cervical fascia from superficial to deep cervical areas. Esophageal rupture, a rare complication of cervical surgery, presents with varying symptoms depending on its location and was particularly challenging to diagnose in this patient due to high cervical tetraplegia, which masked typical pain responses. Therefore, this case highlights the need to consider esophageal rupture in differential diagnoses for chronic ACDF patients, even when typical symptoms are absent.

A 65-year-old patient presented a fever that began three weeks prior. He had a complete spinal cord injury below the C4 level 20 years ago in an accident and underwent anterior cervical discectomy and fusion (ACDF) on C4-5 due to a fracture and dislocation of the cervical 4–5 vertebrae. This patient was diagnosed with a urinary tract infection at another hospital and treated with antibiotics. Urinary tract infections are commonly seen in patients with neurogenic bladders in spinal cord injury. Despite treatment, the fever persisted. The patient was referred to our hospital for further evaluation and treatment due to the persistent fever. Upon examination at our hospital, there was a mass-like lesion in the posterior neck. The inflammatory markers were still elevated. We performed cervical spine MRI scans, which revealed an anterior epidural space abscess and inter-intramuscular abscess involving the left posterior neck area, extending from C3 to C6 (Figure 1). In addition to the soft tissue infection, there was also osteomyelitis in the lamina and transverse processes of the cervical spine, for which vancomycin and a taxolactam (piperacillin sodium and tazobactam) were administered. To address the abscess, a 10 cm incision was made in the left posterior cervical area, allowing for massive irrigation and abscess removal, followed by a curettage of the cervical posterior bony arch. After the abscess removal and bony curettage, primary closure was performed. The duration of antibiotic treatment was anticipated to be between 4 and 8 weeks, depending on the clinical symptoms and the results of the inflammatory markers.

After three months of antibiotic treatment, the patient was discharged due to the improvement of the infection and abscess. However, the patient was readmitted a month later with symptoms of a systemic infection and the formation of an abscess on the posterolateral side once again. To manage the pus drainage, an incision and open drainage were performed. During the subsequent course of antibiotic treatment, food particles were observed in the previously excised drainage pathway. Furthermore, an endoscopic examination revealed an esophageal rupture with protruding hardware of the cervical fusion. (Figure 2) Considering the medical risks of operative repair, a percutaneous endoscopic gastrostomy was performed, and there were no further recurrences of infection after that.

Esophageal rupture can occur due to trauma, cervical operations, or during endoscopic examinations [1]. It is a rare complication reported in 0.02% of patients after acute Anterior Cervical Discectomy and Fusion (ACDF) [2]. However, there are no reports of rupture in patients with chronic ACDF. Symptoms of esophageal rupture can vary depending on the location. Lower esophageal rupture may present with chest pain, vomiting, and shortness of breath [3], while upper esophageal rupture may cause neck pain, dysphonia, and hoarseness. However, in the case of our patient with a high-level complete cervical injury, an accurate perception and localization of pain was impossible. Musculoskeletal damage often leads to increased spasticity or neuropathic pain in areas where sensation is absent. Clinicians must, therefore, be cautious to detect systemic or localized pathologies the patient might not perceive.

Imaging findings of esophageal rupture typically include pleural effusion, pneumothorax, and subcutaneous emphysema [4].Commonly, chest X-ray or chest CT scans reveal mediastinal or pleural symptoms. In the case of our patient, the chest X-ray showed no specific findings, and signs of infection were primarily in the posterolateral neck area without any symptoms of subcutaneous emphysema. The infection occurred deep within the cervical spine without any skin infection, which is different from the typical pattern of spinal osteomyelitis that usually involves the endplate and intervertebral disc first. We speculated that the infection spread to the epidural space and cervical muscles through the cervical fascia. Therefore, although esophageal rupture is rarely reported in chronic ACDF patients and is challenging to detect solely through imaging, it is crucial to consider this condition in the differential diagnosis of atypical deep cervical soft tissue infections. 

## Figures and Tables

**Figure 1 diagnostics-14-00391-f001:**
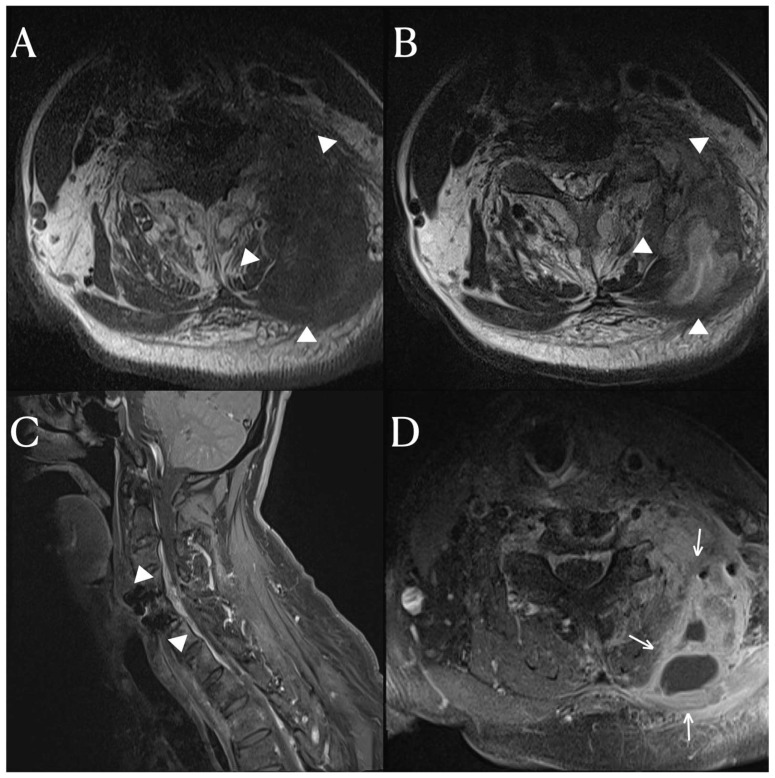
The cervical spinal MRI images, T1 (**A**) and T2 (**B**) weighted axial scans revealed an inter-intramuscular abscess (arrowhead) involving the left scalene, multifidus, semispinalis cervicis, and semispinalis capitis muscles, as well as the levator scapulae. The T1-weighted enhanced axial scan (**D**) displayed a rim-enhanced fluid collection (arrow) indicative of an abscess. The T1-weighted sagittal scan (**C**) demonstrated an epidural abscess (arrowhead) extending from the C3 to C6 levels.

**Figure 2 diagnostics-14-00391-f002:**
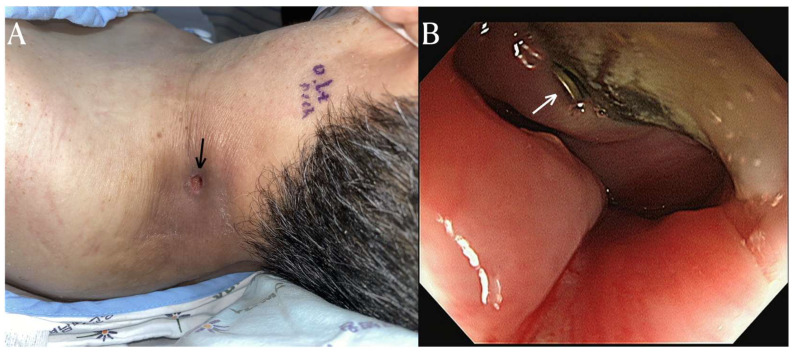
There was a drainage pathway (black arrow) on the posterolateral side of the left neck for the cervical abscess. (**A**) On the endoscopic examination, a metallic device (white arrow) was observed protruding into the esophagus (**B**).

## Data Availability

Data are available on a reasonable request from the authors.

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
