# Peer review of "Diagnostic and Management Challenges of Esophageal Rupture with Concomitant Cervical Abscess in Chronic High Cervical Tetraplegia"

_diagnostics, 2024, doi:10.3390/diagnostics14040391_

Round 1

Reviewer 1 Report

Comments and Suggestions for Authors

It is an interesting case, we must always know this pathology and how to manage these patients.

Well developed paper

Obviously it is not a work that presents a very exceptional case, but it is interesting to know the complications that can arise in quadriplegic patients who have undergone cervical spine surgery and in whom a prosthesis has been placed.

These patients are difficult to manage, and head and neck surgeons rarely participate in their management.

There is little published material on the subject, and it is true that the authors provide very little bibliography. It would surely be positive if the bibliography provided was more extensive, but I repeat that there are few similar works.

The posterior location of the abscess is surprising, that it had spontaneous drainage and that it did not cause other types of complications such as mediastinitis (if it had been located elsewhere in the neck).

I do not have the capacity to assess whether the work is well or poorly written, since my knowledge of English is not sufficient, in any case it seemed sufficient to me.

The conclusions presented are sufficient, the case does not go beyond itself.

The only thing I would appreciate is if the MRI images presented were of better quality, and indicated with arrows the areas to which they refer.

Author Response

We greatly appreciate your time and effort to review our manuscript and provide valuable feedback.

We are pleased that you found the paper well-developed and the case interesting. We acknowledge your point that the case may not be sporadic, but we agree that it is crucial to share knowledge about the management of complications in quadriplegic patients with cervical spine surgery and prostheses.

It is an interesting case, we must always know this pathology and how to manage these patients.

Well developed paper

Obviously it is not a work that presents a very exceptional case, but it is interesting to know the complications that can arise in quadriplegic patients who have undergone cervical spine surgery and in whom a prosthesis has been placed.

These patients are difficult to manage, and head and neck surgeons rarely participate in their management.

There is little published material on the subject, and it is true that the authors provide very little bibliography. It would surely be positive if the bibliography provided was more extensive, but I repeat that there are few similar works.

response:  The scarcity of reports on esophageal rupture in chronic cervical fusion patients is thought to enhance the significance of this paper. Therefore, while the lack of an extensive bibliography is regrettable, paradoxically, the publication of this report is believed to hold high clinical importance.

The posterior location of the abscess is surprising, that it had spontaneous drainage and that it did not cause other types of complications such as mediastinitis (if it had been located elsewhere in the neck).

I do not have the capacity to assess whether the work is well or poorly written, since my knowledge of English is not sufficient, in any case it seemed sufficient to me.

The conclusions presented are sufficient, the case does not go beyond itself.

 The only thing I would appreciate is if the MRI images presented were of better quality, and indicated with arrows the areas to which they refer.

response:  As to the quality of the MRI images, we appreciate your feedback. We will undoubtedly revise these images to improve their clarity and annotate them with arrows to highlight the areas of interest better. This will ensure readers can more easily correlate the imaging findings with the case description.

Reviewer 2 Report

Comments and Suggestions for Authors

The described case has real practical interesе for differential diagnosis and treatment of neck masses and soft tissue inflammation, espeсially in such group of patient with hidden manifestation, but some treatment and surgical options were not shown.

It must be described more clearly about surgical treatment of a neck abscess on the first time and after abscess recurrence, the time of neck fistula occurence and closure.

More detailed literature analysis could improve work quality and significance of aimed conclusions.

Comments on the Quality of English Language

"A 65-year-old patient presented a fever that began three weeks prior. He had a complete spinal cord injury below the C4 level at 20 years ago in an accident and underwent anterior cervical diosectomy (discectomy?) and fusion (ACDF) on C4-5 due to a fracture and dislocation of the cervical 4-5 vertebrae."

 "The patient was referred to our hospital for further 31 evaluation and treatment due to the persistnet ferve (persistent fever?)"

Author Response

Thank you for your insightful review and the constructive comments on our manuscript. We value your feedback, which we believe will significantly enhance the quality and impact of our work.

We agree that the case presented carries practical interest for the differential diagnosis and treatment of neck masses and soft tissue inflammation, particularly in patients with atypical presentations. We understand your concern regarding the omission of some treatment and surgical options, and we will ensure that we include a more comprehensive discussion of these aspects, particularly focusing on the initial and subsequent management strategies for neck abscesses.

The described case has real practical interest for differential diagnosis and treatment of neck masses and soft tissue inflammation, espeсially in such group of patient with hidden manifestation, but some treatment and surgical options were not shown.

It must be described more clearly about surgical treatment of a neck abscess on the first time and after abscess recurrence, the time of neck fistula occurence and closure.

Response: As you have pointed out, we have added additional details regarding the surgical treatment of the neck abscess on lines 36-42 and 44-50 to more clearly address the procedures performed during the initial treatment and after the recurrence of the abscess.

More detailed literature analysis could improve work quality and significance of aimed conclusions.

Comments on the Quality of English Language

"A 65-year-old patient presented a fever that began three weeks prior. He had a complete spinal cord injury below the C4 level at 20 years ago in an accident and underwent anterior cervical diosectomy (discectomy?) and fusion (ACDF) on C4-5 due to a fracture and dislocation of the cervical 4-5 vertebrae."

 "The patient was referred to our hospital for further 31 evaluation and treatment due to the persistnet ferve (persistent fever?)"

response: Thank you for bringing these typographical errors to our attention. We apologize for these oversights and appreciate the opportunity to correct them.

We have corrected the misspelled words "diosectomy" to "discectomy" and "persistnet ferve" to "persistent fever." These errors were inadvertent, and we regret any confusion they may have caused.

Ensuring the accuracy of our manuscript is of utmost importance to us, and we are grateful for your diligence in helping us improve the quality of our work.

Round 2

Reviewer 2 Report

Comments and Suggestions for Authors

Generally, some mistakes and undescribed moments were corrected. This paper has some practical interest in differential diagnosis of neck inflammatory diseases for head and neck and neurosurgeons, due to case rarity, despite the fact that it is presented by non-surgical department specialists.